# When Learning Is Out of Reach, Reset: Generalization in Autonomous Visuomotor Reinforcement Learning

**Zichen Zhang**[†], **Luca Weihs**[†]
[†]PRIOR @ Allen Institute for AI
https://zcczhang.github.io/rmrl

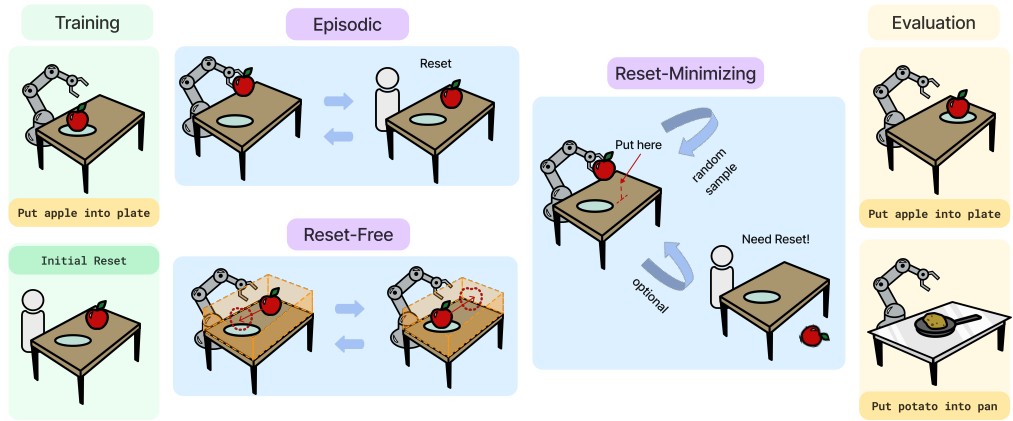

Figure 1: **Episodic, Reset-Free, and Reset-Minimizing RL.** In standard (*i.e.* episodic) reinforcement learning (RL) agents have their environments reset after every success or failure, an expensive operation in the real world. In Reset-Free RL (RF-RL), researchers have designed "reset games" which allow for learning so long as special care is taken to avoid irreversible transitions (*e.g.* an apple falling out of reach). We consider Reset-Minimizing RL (RM-RL) where in realistic and dynamic environments agents may request human interventions but should minimize these requests.

## 1 Introduction

Episodic training, where an agent's environment is reset to some initial condition after every success or failure, is the de facto standard when training embodied reinforcement learning (RL) agents. Work in learning without any resets, i.e. Reset-Free RL (RF-RL) [1, 2, 3, 4, 5, 6, 7, 8, 9, 10], is very promising but is plagued by the problem of irreversible transitions which hinders learning. Moreover, the limited state diversity and instrument setup encountered during RF-RL means that works studying RF-RL largely do not require their models to generalize to new environments.

In this work, we instead look to minimize, rather than completely eliminate, resets while building visual agents that can meaningfully generalize. Refer to Fig 1 for comparisons with episodic, RF-RL, and our proposed Reset-Minimizing RL (RM-RL). We propose a new Stretch Pick-and-Place (STRETCH-P&P) benchmark designed for evaluating generalizations across goals, cosmetic variations, and structural changes. Moreover, towards building performant reset-minimizing RL agents, we propose unsupervised metrics to detect irreversible transitions and a single-policy training mechanism to enable generalization. Our proposed approach significantly outperforms prior episodic, reset-free, and reset-minimizing approaches achieving higher success rates with fewer resets in STRETCH-P&P and another popular RF-RL benchmark. Finally, we find that our proposed approach can dramatically reduce the number of resets required for training other embodied tasks, in particular for RoboTHOR ObjectNav we obtain higher success rates than episodic approaches using 99.97% fewer resets. The full paper is available at https://arxiv.org/abs/2303.17600.

7th Conference on Robot Learning (CoRL 2023), Atlanta, USA.

|  | (a) **Sawyer Peg** | | | (b) **STRETCH-P&P** | | | | | (c) **ObjectNav** | | |
|---|---|---|---|---|---|---|---|---|---|---|---|
|  | IND | OOD | Resets | POS | VIS | OBJ | ALL | Resets |  | Success | SPL | Resets |
| Ours | 1.00±0.00 | 0.98±0.02 | 113.3±7.5 | 0.93±0.07 | 0.78±0.02 | 0.07 | 0.02 | 678.7±40 | Ours | 0.551 | 0.275 | 635 |
| FB-RL+GT | 1.00±0.00 | 0.86±0.08 | 304.0±0.0 | 0.29±0.02 | 0.20±0.00 | 0.02 | 0.007 | 890±236.5 | $H$=300 | 0.355 | 0.167 | 1M |
| Periodic | 1.00±0.00 | 0.98±0.02 | 303.3±1.0 | 0.05±0.04 | 0.04±0.04 | 0.01 | 0.00 | 592.0±0.0 | $H$=10k | 0.418 | 0.218 | 10k |
| Episodic | 1.00±0.00 | 0.95±0.00 | 30k | 0.54±0.04 | 0.36±0.02 | 0.03 | 0.00 | 66k | $H$=∞ | 0.339 | 0.178 | 60 |
| [10] | 0.80±0.20 | 0.05±0.05 | 128.8±44.0 |  |  |  |  |  | [11] | 0.504 | 0.234 | 2M |

Table 1: *Sawyer Peg* for in-domain and out-of-domain random box hole tests within $3M$ training steps, with an extra comparison with supervised method [10]. **STRETCH-P&P** for four evaluations when budget=1, trained for $3M$ steps. *ObjectNav* Benchmark success and SPL evaluating in the unseen validation set, training for $100M$ steps.

## 2 The Stretch Pick-and-Place Benchmark

We build our benchmark within AI2-THOR [12], a high visual fidelity simulator of indoor environments. During evaluation in STRETCH-P&P, a Stretch RE1 Robot is placed before a table within a room. On this table are two objects, a container and a small household item. The agent is given a text description of a task involving how the household item should be moved where this instruction can be semantic *e.g.*, "Put apple into plate", or point-based, "Put the apple at $X$" where $X$ encodes the relative position between the goal coordinate and the agent's gripper. To study generalization, we consider four evaluation settings: (1) Positional out-of-domain (POS-OOD): the environment and objects are seen during training but object and goal positions are randomized to be much more diverse. (2) Visually out-of-domain (VIS-OOD): object instances are the same as in training but the lighting and the materials/colors of background objects will be varied. (3) Novel objects (OBJ-OOD): none of the above visual augmentations will be applied but the container and household object instances will be distinct from those seen during training. (4) All out-of-domain (ALL-OOD): the agent experiences visual augmentations from (2), novel object instances as in (3), and the addition of new background distractor objects simultaneously.

During training, the agent is placed before a table with a container and a household object. The table, lighting, and object materials are all kept constant during training. Upon requesting a reset, the agent's position, as well as the position of the two objects, may be placed into any initial configuration. As achieving this generalization may be challenging, we do consider allowing more diversity to be introduced during training by allocating a budget for more than one seen object or container.

## 3 Methods

**Measures of Irreversibility.** Some irreversible transitions are explicit, e.g. a glass is dropped and shatters. However, in a more complex real-world environment, they may be more subtle. In such cases, the robot may find success challenging, but not strictly impossible. We refer to these states that are difficult, but not impossible, to recover from as near-irreversible (NI) states. Intuitively, undergoing an NI transition should correspond to a decrease in the degrees of freedom available to the agent to manipulate its environment: that is, if an agent underwent an NI transition at timestep $i$ then the diversity of states $\tau_\pi(i+1), \ldots, \tau_\pi(t)$ should be small compared to the diversity before undergoing the irreversible transition. To formalize this, we can compute the above count, which we call $\varphi_{W,\alpha,d,}(\mathcal{T}_t)$, as

$$\max_{(i_0,\ldots,i_m)\in P(t)} \sum_{j=0}^{m-1} 1_{[i_{j+1}-i_j \geq N]} \cdot 1_{\{d(\tau_\pi(i_j),\ldots,\tau_\pi(i_{j+1}-1))<\alpha\}}$$

where $d : \mathcal{S}^H \to \mathbb{R}_{\geq 0}$ is some non-negative measure of diversity among states. As $\varphi_{W,\alpha,d}$ is a counting function, we can turn it into a decision function simply by picking some count $N > 0$ and deciding to reset when $\varphi_{W,\alpha,d} \geq N$. In our experiments we evaluate several diversity measures $d(s_1, \ldots, s_H)$ including: (1) a dispersion-based method using an empirical measure of entropy or the mean standard deviation of all $s_i$, (2) a distance-based method using Euclidean distance or dynamic time warping (DTW).

**Single Policy.** In contrast to the multi-policy Forward-Backward RL (FB-RL) approaches used by most works studying FB-RL, we aim to use a single policy to achieve RM-RL that can adapt to general embodied tasks. Recall the objective for goal-conditioned POMDP in traditional episodic RL: $\arg\max_\pi \mathbb{E}\left[\sum_{t=0}^{\infty} \gamma^t r(s_t, a_t \mid g)\right]$. In FB-RL, the "forward" goal space is normally defined as a singleton $\mathcal{G}_f = \{g^\star\}$ for the target task goal $g^\star$ (e.g. the apple is on the plate). The goal space for "backward" phase is then the (generally limited) initial state space $\mathcal{G}_b = \mathcal{I} \subset \mathcal{S}$ such that $\mathcal{G}_f \cap \mathcal{G}_b = \emptyset$. As the goal spaces in FB-RL are disjoint and asymmetric, it is standard for separate forward/backward policies and even different learning objectives for training FB-RL agents. In our setting, however, there is only a single goal space which, in principle, equals the entire state space excluding the states we detect as being NI states.

## 4 Experiments

We consider three tasks in different embodied settings: the STRETCH-P&P, Sawyer Peg [13, 2], and RoboTHOR ObjectNav [14] tasks. [1] We compare against FB-RL with ground truth resets for explicit irreversible states (FB-RL+GT) and a periodically resetting approach which resets every fixed number, *e.g.* 10k, of steps (Periodic). All models are trained using the PPO [15] RL algorithm. We use frozen CLIP [16] with CNN adapters to encourage visual generalization and language understanding for STRETCH-P&P tasks. The model used for Sawyer Peg is similar as [17, 18] but we only use *single* CNN visual encoder that digests both views for parameter-efficiency. We use the same ResNet50 CLIP architecture with only egocentric visual observation input for ObjectNav as proposed in [11].

As shown in Table 1, our method achieves high success rates more consistently and with far fewer resets than other baselines. Surprisingly our method is also more efficient in terms of training steps. This suggests that our measures of NI transitions can consistently and accurately identify time-points where a reset will be of high value for learning. Intuitively the forward-backward gameplay of FB-RL models should be easier to learn than when using random targets as the space of goal states of FB-RL is a small subset of those used when randomizing targets. However, we demonstrate that random targets introduce little additional difficulty over FB-RL, and provide significant benefit in positional generalizations. More details can be found at https://zcczhang.github.io/rmrl.

---

[1]Visualizations at https://zcczhang.github.io/rmrl

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
