# OpenReview forum: "When Learning Is Out of Reach, Reset: Generalization in Autonomous Visuomotor Reinforcement Learning"
_robot-learning.org/CoRL/2023/Workshop/OOD — OOD Workshop @ CoRL 2023_

### Official Review · Reviewer_cUbi · 2023-10-16

**Rating:** 8
**Confidence:** 4

**Review:**

Summary:

This paper presents an approach to train generalizable manipulator policies operating on visual inputs. A key challenge when training manipulator is that certain actions may be (near)-irreversible, and thus either preclude the agent from adequately exploring the state space, or require human intervention during training to reset the system. In this work, the authors propose a strategy for minimizing human intervention by proposing an unsupervised method for detecting transitions into near-irreversible states. In addition, they propose a benchmark for evaluating how RL manipulators can generalize to various forms of OOD scenarios. They show that this method allows training a policy that generalizes better than prior work as well as ablations against alternative resetting strategies.

Strengths:

- The paper tackles an important problem of training RL agents that can generalize without the often expensive restriction of having humans reset the robot environment very often.
- The strategy offers compelling performance gains relative to the baselines and ablations, achieving higher success rates than a fully episodic RL baseline with only a fraction of the number of resets required.
- The Stretch-P&P benchmark proposed offers a strong experimental testbed to evaluate how visuomotor manipulators generalize to scenes that are OOD in varying ways.

Weaknesses:

- The paper was hard to follow at times. I would have appreciated if there was more discussion explaining the measure determining if a near-irreversible transition had occurred at a more intuitive level.
- The paper would be improved with more discussion and analysis of the results. While it is clear why the proposed method might achieve similar or better in-distribution performance with fewer resets as compared to episodic RL, I'm curious if you have any theories as to why the proposed method also generalizes better in OOD scenarios? Does reset-minimizing RL encounter a greater diversity of states than periodic resetting or episodic RL?

---

### Official Review · Reviewer_cf2G · 2023-10-16
**Review of RM-RL**

**Rating:** 5
**Confidence:** 4

**Review:**

**Summary:** This paper proposes a reset-minimizing reinforcement learning (RM-RL) framework to reduce the cost of resets in the standard RL setup as well as address the limitations of reset-free RL methods. Specifically, the paper introduces a function that quantifies the diversity of states along a trajectory, which is ultimately used to decide when a reset is required.

**Relevance:** OOD generalization of learning-based policies is relevant to this workshop and the community. This paper is most relevant to 3) “Episodic interaction with an environment: Can we develop methods that mitigate or account for shifted conditions that consistently degrade a learning-enabled robot's performance?”

**Novelty:** There is some novelty to this paper, for example, the notion of using state diversity to determine reset points for RL.

**Significance:** In two of the three domains, the proposed method is demonstrated to either 1) outperform reset-free RL in OOD scenes or 2) achieve similar performance with fewer resets. For robot learning in the real-world, the result of 2) holds significance.

**Quality / clarity:** The quality and clarity of the paper is subject to improvement. For example, it is very difficult to comprehend what the core component of the methodology described in Section 3 (Measures of Irreversibility) is doing, largely because mathematical symbols are not introduced before or after they are used. Moreover, precisely how the counting function is used within the framework is unclear.

**Suggestions:** Much emphasis was placed describing the Stretch-P&P dataset in Section 2. It would be helpful to also comment on why the other benchmarks were included in the evaluation.

---

### Decision · Program_Chairs · 2023-10-17

**Decision:**

Accept

**Comment:**

We agree with the reviewers’ assessment that this work is technically sound and will contribute to productive, topical discussions at the 2023 Workshop on OOD Generalization in Robotics. In particular, we appreciate that this work proposes a new benchmark for generalization performance; the development and standardization of this and other such benchmarks is a key desired outcome of this workshop. We recommend the authors incorporate the reviewers’ feedback into their camera-ready submission to further improve their manuscript.